# Immunomodulatory Roles of PARP-1 and PARP-2: Impact on PARP-Centered Cancer Therapies

**DOI:** 10.3390/cancers12020392

**Published:** 2020-02-08

**Authors:** José Yélamos, Lucia Moreno-Lama, Jaime Jimeno, Syed O. Ali

**Affiliations:** 1Cancer Research Program, Hospital del Mar Medical Research Institute (IMIM), 08003 Barcelona, Spain; morenolamalucia@gmail.com; 2Immunology Unit, Department of Pathology, Hospital del Mar, 08003 Barcelona, Spain; 3Department of General Surgery, Breast Unit, Hospital Universitario Marqués de Valdecilla, 39008 Santander, Spain; jaime.jimeno@scsalud.es; 4Oxford University Hospitals, NHS, Oxford OX3 9DU, UK; omar.ali1795@gmail.com

**Keywords:** PARP, immunomodulation, tumor microenvironment

## Abstract

Poly(ADP-ribose) polymerase-1 (PARP-1) and PARP-2 are enzymes which post-translationally modify proteins through poly(ADP-ribosyl)ation (PARylation)—the transfer of ADP-ribose chains onto amino acid residues—with a resultant modulation of protein function. Many targets of PARP-1/2-dependent PARylation are involved in the DNA damage response and hence, the loss of these proteins disrupts a wide range of biological processes, from DNA repair and epigenetics to telomere and centromere regulation. The central role of these PARPs in DNA metabolism in cancer cells has led to the development of PARP inhibitors as new cancer therapeutics, both as adjuvant treatment potentiating chemo-, radio-, and immuno-therapies and as monotherapy exploiting cancer-specific defects in DNA repair. However, a cancer is not just made up of cancer cells and the tumor microenvironment also includes multiple other cell types, particularly stromal and immune cells. Interactions between these cells—cancerous and non-cancerous—are known to either favor or limit tumorigenesis. In recent years, an important role of PARP-1 and PARP-2 has been demonstrated in different aspects of the immune response, modulating both the innate and adaptive immune system. It is now emerging that PARP-1 and PARP-2 may not only impact cancer cell biology, but also modulate the anti-tumor immune response. Understanding the immunomodulatory roles of PARP-1 and PARP-2 may provide invaluable clues to the rational development of more selective PARP-centered therapies which target both the cancer and its microenvironment.

## 1. Introduction

Poly(ADP-ribose) polymerase-1 (PARP-1) and PARP-2 are two enzymes of the PARP family of proteins that, in response to DNA damage, catalytically cleave β-NAD^+^ and transfer ADP-ribose moieties onto specific amino residues of acceptor proteins. This process, termed poly(ADP-ribosyl)ation (PARylation), forms poly(ADP-ribose) (PAR) polymers varying in size and branching, which have diverse functional and structural effects on target proteins [1,2,3]. The deletion of either PARP-1 or PARP-2 in mice is associated with disturbances of DNA integrity and repair, supporting key shared functions of these proteins that are pivotal to DNA repair [4]. Indeed, combined PARP-1 and PARP-2 deficiency leads to embryonic lethality [5], which is likely due to their central role in the DNA damage response (DDR) [2,4].

Studies based on the role of these PARPs in the DDR in cancer cells have led to the development of PARP inhibitors as new therapeutic tools in cancer, both as adjuvant treatment potentiating chemotherapy, radiotherapy, and immunotherapy and as monotherapy exploiting cancer cell-specific defects in DNA repair, such as BRCA mutations [6,7,8,9]. However, the tumor microenvironment is formed from more than just tumor cells, and also includes stromal cells and infiltrating cells of the innate and adaptive immune system, which are likely to also be affected by PARP inhibition. These cells communicate with each other through direct contact and/or indirect signals that can alter the functionality of immune cells so that they either favor or limit tumor growth [10,11]. Emerging evidence supporting the immunomodulatory roles of PARP-1 and PARP-2 has raised the prospect of harnessing PARP inhibition to not only target the cancer itself, but also therapeutically modify its microenvironment.

In this review, we highlight the functions of PARP-1 and PARP-2 in the immune system and how their immunomodulatory roles might impact the response to tumors. We will examine recent data suggesting specific and redundant roles of PARP-1 and PARP-2 in the innate and adaptive immune responses and the immunological potential of PARP inhibitors. Understanding the immunomodulatory roles of PARP-1 and PARP-2 may provide invaluable clues for the rational development and exploitation of more selective anti-cancer PARP inhibitor drugs, both as new monotherapeutic approaches and in combinations with immunotherapy.

## 2. Impact of PARP-1 and PARP-2 on T Cell Development and Function

T cell development is a highly regulated process beginning in the thymus from bone marrow-derived lymphoid precursors, and giving rise to mature T cells through well-characterized sequential maturation steps involving a complex transcriptional network orchestrating cell proliferation, survival, and differentiation [12]. The earliest thymic progenitors are named double-negative (DN) cells, comprising four fractions (DN1 to DN4), which are characterized by a lack of CD4 and CD8 surface markers. DN2 and DN3 thymocytes express recombination-activating genes (Rag) and undergo extensive T cell receptor (TCR) β, γ, and δ gene rearrangement to express functional TCR chains. A successful recombination of TCRγ and TCRδ promotes the generation of γδ T cells. In contrast, the generation of αβ T cells requires additional differentiation steps. A successfully rearranged TCRβ chain associates with CD3 chains to form a pre-TCR. The expression of a pre-TCR drives DN4 differentiation into double-positive (DP) thymocytes—the most abundant population in the thymus—expressing both CD4 and CD8 surface markers. During this stage of development, the thymocytes re-express the Rag genes, which allows multiple rounds of TCRα gene rearrangements to increase the probability of forming a functional αβ TCR. DP thymocytes undergo a very strict selection process, such that those that express a TCR which is not able to interact with self-major histocompatibility complex (MHC)/self-peptide complexes die due to neglect. In the same way, the DP thymocytes that bind self-MHC/self-peptide molecules with a high affinity are eliminated by negative selection. Meanwhile, those DP thymocytes expressing TCRs that bind self-MHC/self-peptide ligands with a low affinity are positively selected and differentiated into either CD4^+^ or CD8^+^ single-positive (SP) thymocytes [12]. At this stage of development, some CD4^+^ thymocytes express the transcription factor Forkhead box protein 3 (FoxP3), which confers the cells an immunosuppressive function (Treg) [13]. All kinds of T cells generated in the thymus will seed the peripheral lymphoid tissues (Figure 1).

Although both PARP-1 and PARP-2 proteins are expressed in thymocytes [14], only PARP-2 plays a significant role in thymocyte development. Therefore, PARP-2-deficient, but not PARP-1-deficient, mice show a significant reduction in the number of DP thymocytes. This phenotype is associated with the role of PARP-2 in preventing the accumulation of DNA double-strand breaks (DSBs) and the resulting activation of a DNA damage-induced apoptotic response during TCRα rearrangements [14,15]. In fact, p53 deficiency restores thymocyte populations in PARP-2-deficient mice [15]. In contrast, PARP-1 regulates tTreg development [16], while PARP-2 does not seem to play any role in tTreg development [17] (Figure 1).

Once the T cells in the thymus have matured, they migrate to the peripheral lymphoid tissues forming the naïve T cell pool, where they continue their differentiation to become fully immunocompetent to mount appropriate immune T cell responses to antigen challenge [18]. Naïve T cells proliferate both in situations of lymphopenia (homeostatic proliferation) driven by TCR/self-peptide–MHC interactions, and in response to antigen challenges driven by TCR/foreign-peptide–MHC interactions and co-stimulation, accompanied by differentiation into effector T cells (Th1, Th2, Th17, Treg, and cytotoxic T cells) and the final generation of a memory T cell population [19] (Figure 1). The control of T cell homeostasis is not only mediated by MHC–TCR interactions and cytokine-mediated signals, but also processes which regulate essential T cell functions to maintain genomic stability, such as cell-cycle checkpoints, DNA repair, and apoptosis [20,21]. 

Although PARP-1 deficiency or PARP-2 deficiency alone does not affect the number of T cells in peripheral lymphoid tissues [14,17], double deficiency in the T cell compartment results in a significant decrease in both CD4^+^ and CD8^+^ peripheral T cells [17]. The T cell lymphopenia present in mice with double PARP-1 and PARP-2 deficiency indicates that these proteins act in a coordinated manner to prevent the accumulation of unrepaired DNA breaks upon homeostatic proliferation or in response to antigen challenge, but not under basal conditions, avoiding T cell death [17]. PARP-1 and PARP-2 likely act through the principle of synthetic lethality [22], whereby they regulate two independent, but functionally linked, processes. T cell lymphopenia in double-deficient mice for PARP-1 and PARP-2 blunts the anti-viral immune response and the response to other T cell-dependent antigens [17]. Furthermore, double deficiency of PARP-1 and PARP-2 in the T cell compartment in mice affects the T cell response to tumors [23]. Although PARP inhibitors do not achieve the persistence of inhibition as obtained in T lymphocytes with double genetic deficiency of PARP-1 and PARP-2, these pharmacological inhibitors can still impact the T cell compartment and thus the T cell immune response. Indeed, in a mouse breast tumor model induced by the AT-3 cell line, which is sensitive to the PARP inhibitor olaparib, the anti-tumor effect of olaparib is blunted by an intact immune system [23]. As such, it would be interesting to study how PARP inhibitors used in the clinic affect the immune compartment in patients.

Transcriptional activation via different signaling pathways is fundamental to the differentiation of T cells. Among the key transcription factors in T cell development and function are the nuclear factor of activated T cells (NFAT) family of transcription factors (NFAT1 to NFAT5) [24]. After antigenic recognition by the TCR, a signaling cascade is initiated in the T cell, leading to the activation and nuclear translocation of NFAT1, NFAT2, and NFAT4, where, in combination with other transcription factors such as AP1, they regulate the expression of cytokines and lineage-specific transcription factors to control pathways of T cell differentiation into Th1 or Th2 types [24]. Of note, our group has demonstrated that PARP-1 is activated during T cell activation, where it modulates the activity of NFAT through PARylation, as evidenced by PARP inhibitors causing an increase in NFAT-dependent transactivation [25]. Moreover, PARP-1 plays a critical role in the gene expression reprogramming that takes place in T cells upon activation [26]. Indeed, PARP-1-deficiency seems to bias the T cell response to a Th1 phenotype [26] and has been shown to reduce differentiation into Th2 cells in different experimental models [26,27]. The pharmacological inhibition of PARP has led to more controversial results, where, in one case, the PARP inhibitor led to an increase in Th1 cytokine production and a reduction in Th2 cytokines [28], while in another case, the inhibition led to a decrease in Th1 cells [29]. These discrepancies may be associated with the type of inhibitor used or the experimental model.

PARP-1 also plays a role in the generation of Treg cells in the periphery (pTreg) from CD4^+^ T cells that express FoxP3, and PARP-1-deficient mice have been found to display an increased number of Treg cells [16]. Moreover, PARP-1 negatively regulates the suppressive function of Treg cells at the posttranslational level through FoxP3 PARylation [30]. PARP-1 can also regulate the generation of Treg cells through its role in regulating the expression of transforming growth factor β receptors (TGFβR) in CD4^+^ T cells, and therefore affects TGFβ signaling in T cells [30]. Interestingly, the inhibition of TGFβRI expression by PARP-1 is dependent on PARP enzymatic activity, while the inhibition of TGFβRII expression depends on the interaction of PARP-1 with the promoter of the TGFβRII gene [31]. In contrast, the function of PARP-2 in transcriptional regulation in T cells remains unclear.

## 3. Impact of PARP-1 and PARP-2 on B cell Development and Function

As with the development of T lymphocytes, the development of B cells, which takes place in the bone marrow, is also a precisely regulated process that starts from pluripotent hematopoietic stem cells. In the first step, the hematopoietic stem cells differentiate into pro-B cells that transiently express the Rag genes, which mediate immunoglobulin (Ig) heavy-chain gene rearrangements to assemble a V_H_DJ_H_-Cμ protein. The association of this protein with Vpre-B and λ5 surrogate light chain proteins leads to the formation of the pre-B cell receptor (BCR) complex in the large pre-B cell population. Pre-BCR signaling results in differentiation into small pre-B cells, which re-express the Rag genes, allowing Ig light chain V_L_J_L_ gene rearrangement. The succeeding association of V_L_J_L_ with V_H_DJ_H_-Cμ generates fully functional membrane-bound IgM receptors in immature B cells, which further differentiate into transitional B cells which co-express IgM and IgD receptors on their surface [32] (Figure 2). Despite the role of PARP-1 and PARP-2 in DNA repair, their role in Ig V(D)J gene recombination has remained unclear or unknown [33,34,35]. Recent data obtained by our group show that mice with dual, but not individual, PARP-1 and PARP-2 deficiency exhibit a reduced number of B cells in the bone marrow [36] (Figure 2). A possible explanation for this bone marrow hypocellularity is that the V(D)J recombination process is defective in these cells. However, a detailed analysis showed that neither single nor double PARP-1/PARP-2 deficiency affected Ig V(D)J gene recombination [36]. As in the T cell compartment, B cell lymphopenia in dually PARP-1- and PARP-2-deficient mice is associated with an accumulation of unrepaired DNA damage in proliferating B cells leading to cell death, suggesting a potential model whereby coordinated signals from PARP-1 and PARP-2 are required to maintain genomic integrity during lymphoid proliferation. This is consistent with recent data showing that dual PARP-1 and PARP-2 deficiency results in the accumulation of replication-associated DNA damage due to the impaired stabilization of Rad51 at damaged DNA replication forks and uncontrolled DNA resection thereafter [37].

The differentiation of transitional B cells leaving the bone marrow continues in the peripheral lymphoid organs, which gives rise to mature marginal zone (MZ) or mature follicular (FO) B cells [32]. After their interaction with antigens, mature B cells will either proliferate and differentiate extra follicularly (Bcl-6^−^) or form germinal centres (Bcl-6^+^), ultimately generating short-lived antibody secreting plasmablasts or long-lived memory B cells and antibody-secreting plasma cells, respectively [38,39,40] (Figure 2). Antibody diversity is in part achieved during these responses through DNA editing via both Ig class-switching recombination (CSR) and somatic hypermutation (SHM), which are mediated by activation-induced cytidine deaminase (AID) [41].

While single PARP-1 or PARP-2 deficiency does not affect the B cell compartment in peripheral lymphoid tissues [14,17], combined PARP-1 and PARP-2 deficiency impairs peripheral B cell homeostasis [36]. This lymphopenia does not affect all B cell populations equally, such that only the number of FO B cells is dramatically reduced in mice with double PARP-1 and PARP-2 deficiency, while the number of MZ B cells is not affected. The reason for this is unclear and requires further exploration [36] (Figure 2). Interestingly, double PARP-1 and PARP-2 deficiency impairs antibody responses to T cell-independent, but not T cell-dependent, antigens [36]. In addition, T cell-independent antigens elicit IgG1- and IgG2b-predominant antibody responses in single PARP-1-deficient mice [36,42] (Figure 2). It is important to note that despite their role in DNA repair, neither PARP-1 nor PARP-2 are required for CSR [36,43] and instead, their role in B cell homeostasis underpins their importance for Ig responses to specific antigens. Another role of the PARPs in B cell development is the role of PARP-1 activation in switching off Bcl6 [44]—a transcription factor essential for the formation of germinal centers [45,46]. However, germinal center formation upon immunization is normal in mice with single or dual deficiencies of PARP-1 and PARP-2 [36]. Meanwhile, the role of PARP-1 in SHM is controversial, with some data showing a dispensable role [47], while other data indicate a role of PARP-1 in SHM [48]. Meanwhile, the role of PARP-2 in SHM is unknown. 

## 4. Role of PARP-1 and PARP-2 in the Cellular Components of the Innate Immune System

In addition to their role during the development and function of cellular components of the adaptive immune system, PARP-1 and PARP-2 have also been involved in different functional aspects of cells involved in the innate immune response, including neutrophils, macrophages, dendritic cells, and natural killer (NK) cells. These innate immune cells serve as the front line of host protection to infection and non-infectious tissue damage. In addition, cells of innate immunity are critical for stimulating subsequent adaptive immune responses [49].

Neutrophils are key players in acute and chronic inflammatory responses through their role in phagocytosis, the recruitment of other immune cells, and the secretion of antibacterial proteins [50]. In cancer, tumor-associated neutrophils are thought to contribute to inflammation in the tumor [51]. Of note, PARP-1 is important in the recruitment and function of neutrophils in different processes related to inflammation [52,53,54,55]. Meanwhile, the role of PARP-2 in neutrophil biology remains elusive (Figure 3).

Macrophages are differentiated from circulating monocytes after extravasation into tissues. Upon differentiation, macrophages are prepared to sense and respond to infection and tissue injuries through the phagocytosis of dead cells, debris, and foreign materials [56]. Besides phagocytosis, macrophages are also important as antigen-presenting cells (APC) to T cells [57]. Macrophages show considerable plasticity, which permits them to adapt their phenotype in response to different microenvironments. There are two major forms of activated macrophages, termed pro-inflammatory M1, which is characterized by the production of pro-inflammatory cytokines, and anti-inflammatory M2, which is characterized by the secretion of anti-inflammatory cytokines [58]. Of note, PARP inhibitors inhibit the expression of LPS-induced proinflammatory cytokines like tumor necrosis factor α (TNFα), interleukin-1 (IL-1), and IL-6 by macrophages [59]. Meanwhile, recent work has shown that PARP-1, but not its enzymatic activity, enhances the transcriptional activity of LPS-induced proinflammatory genes in macrophages [60]. This effect would be mediated by the modulatory role of PARP-1 on the transcription factor NF-κB [61]. In addition, functional interplay between PARP-1 and lysine-specific histone demethylase 1A (LSD1) protects pro-inflammatory M1 macrophages from death under oxidative conditions [62]. Moreover, macrophage recruitment in an airway inflammatory model was severely blocked in PARP-1-deficient mice [63]. Meanwhile, the role of PARP-2 in macrophages remains unknown (Figure 3).

Dendritic cells (DC) are specialized APC which process antigen and present it in the context of self-MHC molecules to T cells. In addition, they also upregulate cell surface receptors, including CD80, CD86, and CD40, which interact with co-receptors on the T cells surface (CD40L and CD28), in order to induce proper T cell activation [64,65]. While the role of PARP-1 in the recruitment of DC to tissues in different pathological situations seems to be well-established, its role in the differentiation and function of these cells is less clear [66,67,68,69]. On the other hand, the function of PARP-2 in DC remains unexplored (Figure 3).

NK cells have a wide array of inhibitory and stimulatory receptors on their cell surface that are used for immune surveillance. Upon activation, NK cells show potent cytolytic activity in response to infected or transformed cells by releasing cytotoxic perforin and granzyme and activating apoptotic pathways in target cells through the production of TNFα or via direct cell–cell contact through activation of the tumor necrosis factor-related apoptosis-inducing ligand (TRAIL) and Fas ligand (FASL) pathways [70,71]. Recent work has demonstrated important roles of PARP-1 in NK cell biology. For instance, PARP-1 controls NK cell recruitment to the site of viral infection [72,73]. In addition, PARP-1 is involved in the downregulation of NK cell-activating receptor ligands for immune evasion in acute myeloid leukemia [74] (Figure 3).

## 5. How Could the Immunomodulatory Roles of PARP-1 and PARP-2 Impact the Immune Response to Tumors?

Tumors contain not only cancer cells, but other cell types, including tissue-resident and peripherally-recruited immune cells, fibroblasts, and endothelial cells, which form the tumor microenvironment. Interactions between these cells—cancerous and non-cancerous—are known to either favor or limit tumorigenesis. Indeed, cancer progression is a dynamic process that, based on those interactions, has been divided into three stages: elimination, equilibrium, and escape [75].

During the cancer elimination phase, a competent immune response takes place in which innate and adaptive immune cells are recruited to the tumor microenvironment, where they exert a strong anti-tumor response [51] (Figure 4). The aforementioned immunomodulatory functions of PARP-1 and PARP-2 would thus be expected to have an impact on the immune response against the tumor. Indeed, we have observed a reduction in tumor growth in PARP-1-deficient host-mice and in PARP-2-deficient host-mice, compared to wild-type specimens, in both a C57 syngeneic tumor model induced by the AT-3 breast tumor cell line [23] and in a Balb/c syngeneic tumor model induced by the LP07 lung adenocarcinoma cell line [76], in which both cancer cells lines are proficient for PARP-1 and PARP-2 proteins. This effect may be associated with their immunomodulatory roles.

T cells, in particular, CD8^+^ cytotoxic T cells (CTL) and CD4^+^ Th1 cells, are major contributors to the adaptive host-defense against tumors [10]. Tumor-derived antigens are processed by APCs (mainly dendritic cells), carried to draining lymph nodes and presented to naïve T cells, in order to prime them. Antigen presentation, together with the induction of co-stimulatory signals mediated by the binding of CD28 on the T cell to CD80/CD86 on the APC, leads to the differentiation of naïve CD8^+^ T cells into tumor-specific CTLs. These, in turn, migrate to the tumor microenvironment to kill cancer cells through the secretion of perforin and granzyme [77]. The anti-tumor effect of CD4^+^ Th1 cells is mediated through the secretion of IL-2, TNFα, and IFNγ, enhancing CD8^+^ T cell responses and activating macrophages and NK cells [78,79,80] (Figure 4). Previous data from our group has indicated that coordinated signals from PARP-1 and PARP-2 are required to maintain T cell homeostasis and for the differentiation from naïve to effector T cells affecting both CD4^+^ and CD8^+^ lineages [17]. Accordingly, T cell lymphopenia in dual PARP-1/PARP-2-deficient mice can affect the recruitment of lymphocytes to the tumor microenvironment [23]. Moreover, a defect in the ability of dually PARP-1/PARP-2-deficient T cells to differentiate into effector cells could have consequences for the anti-tumor response (Figure 4). On the other hand, PARP-1-deficiency seems to bias T cell responses to a Th1 phenotype [26] that may also impact tumor progression. Similarly, B cell lymphopenia in dual PARP-1/PARP-2-deficient mice can affect the recruitment of B cells to the tumor microenvironment (Figure 4).

The aforementioned biological roles of PARP-1 in macrophage biology may impact the response of these cells to tumors. Classically-activated M1 macrophages can kill many tumor cells by mechanisms including the recognition of damage-associated molecular patterns (DAMPs) from dying tumor cells and the production of nitric oxide. In addition, the co-operation of T cells and macrophages through direct contact or through the secretion of cytokines is important in the anti-tumor response [77]. Recent work from Hottiger´s group shows that mice with a conditional loss of PARP-1 in myeloid lineages fail to control tumor growth in an MC-38-induced tumor model of colon cancer, which could be attributed to reduced Th1 and CD8^+^ T cell responses [60], suggesting that PARP-1 in macrophages controls Th1 responses to tumors (Figure 4). However, this intrinsic role of PARP-1 in myeloid cells is independent of its enzymatic activity, so would be of limited utility from the point of view of a pharmacological blockade [60].

Intratumor NK cells have been shown to play a very important role in the control of tumor growth [81]. The balance between stimulatory and inhibitory receptor signals determines the activation of NK cells against tumor cells. NK cells may also be activated to kill tumor cells coated with anti-tumor antibodies by antibody-dependent cell-mediated cytotoxicity (ADCC) (Figure 4). Moreover, the tumoricidal capacity of NK cells is increased by cytokines (IL-2, IL-15, IL-12). Although the role of PARP-1 and PARP-2 in modulating NK cell activity against tumors is largely unknown, the aforementioned function of PARP-1 in controlling NK cell recruitment to the site of viral infection [72,73] and its role in the downregulation of NK cell-activating receptor ligands to evade immune surveillance in acute myeloid leukemia [74], may impact tumor progression. 

Tumor cells that have escaped the immune response undergo different strategies in order to acquire immune tolerance, including (i) tumor cell-intrinsic modifications, like the loss of human leukocyte antigens (HLA) class I molecules, loss of tumor-associated antigens, and increased resistance to cell killing by immune cells, and (ii) the generation of an immuno-suppressive microenvironment through the recruitment of cells with immunosuppressive activities (Treg, macrophages M2, and myeloid-derived suppressive cells); the expression of inhibitory checkpoint molecules such as PD-L1, PD-L2, and cytotoxic T lymphocyte-associated antigen 4 (CTLA4) [82,83,84]; the deprivation of nutrients and oxygen; and the secretion of immunosuppressive cytokines (TGFβ, IL-10, and VEGF) [85] (Figure 4).

The previously mentioned roles of PARP-1 in Treg development and function [16,30] may impact the response to tumors. In addition, PARP-1 inhibition leads to the up-regulation of TGFβ receptor expression in CD4^+^ T cells that subsequently affects TGFβ signal transduction [31], which may impact the response to tumors (Figure 4).

However, these tumor microenvironment escape mechanisms can be modified by different strategies in order to reactivate the immune response against tumor cells [86]. Indeed, re-activating the normal function of immune cells in the tumor microenvironment is one of the biggest challenges in oncology research. Accordingly, emerging immunotherapeutic strategies aim to reverse immune tolerance either by modulating T cell co-receptor signals or boosting the recognition of tumor-associated antigens by using monoclonal antibodies [10]. In addition to those strategies based on biological approaches, modifying the immune response through a small-molecule approach targeting intracellular signaling pathways, such as with PARP inhibitors, may represent a breakthrough that is complementary to, and potentially synergistic with, immunotherapy [87].

## 6. PARP Inhibitors as Immunomodulatory Agents

PARP proteins exert their function through their physical association with or by the PARylation of partner proteins [3]. Although most of the immunomodulatory roles of PARP proteins have been based on studies of mice with the genetic deletion of these proteins (Table 1), PARP inhibitors might induce similar immune cell alterations that will modify their interaction with tumor cells. Indeed, recent work has shown how PARP inhibitors might impact the mechanisms used by tumors to evade immunity, although many of these studies are focused on tumor cell-intrinsic mechanisms. These studies can provide information to rationalize the combined use of PARP inhibitors with other strategies aimed at reactivating the immune system against the tumor.

One of the most successful strategies for reinstating an existing anti-cancer T cell immune response is the use of blocking antibodies against cell surface inhibitory co-receptors like cytotoxic T lymphocyte-associated protein 4 (CTLA4) and programmed cell death 1 (PD-1), which block the engagement of PD-1 or CTLA4 with their ligand (PD-L1 and PD-L2 for PD1; CD80/CD86 for CTLA4), thus avoiding the initiation of signaling pathways leading to the suppression of T cell activation. Of note, PARP inhibitors upregulate the expression of PD-L1 in cancer cells and enhance cancer-associated immunosuppression (Figure 4). This immunosuppression is reversible by blocking the PD-1/PD-L1 interaction [88]. This study established the rationale of combining PARP inhibitors with checkpoint blockade agents [7,89,90,91] or agents that alter PD-L1 expression [92], which has led to numerous clinical trials (Table 2). Although the result of an early phase II clinical trial combining Durvalumab with Olaparib in patients with relapsed small cell lung cancer did not meet the preset bar for efficacy [93], we are awaiting the results of ongoing clinical trials to better judge their effectiveness. Moreover, the PARP inhibitor Niraparib has been shown to enhance type I interferon signaling and T cell infiltration in the tumor and improve the therapeutic effect of anti-PD-1 [94].

NK cells can kill cancer cells by inducing death receptor-mediated apoptosis through the expression of FasL or TRAIL [71]. PARP inhibitors have been shown to sensitize cancer cells to death receptor-mediated apoptosis by upregulating death receptor surface expression [95,96] (Figure 4). In addition, the inhibition of PARP-1 upheld the capacity of NK cells to kill myeloid leukemic cells, and restored the proliferation and cytokine production of NK cells and cytotoxic T cells [97].

Recent work has revealed the intriguing link between genomic instability and the accumulation of DNA in the cytoplasm, which triggers the activation of innate immune responses through the cyclic GMP-AMP synthase (cGAS)/stimulator of interferon genes (STING) pathway that evolved to signal the presence of exogenous DNA [98]. Accordingly, it has been shown that PARP inhibitors promote the accumulation of cytosolic DNA, which activates the DNA-sensing cGAS–STING pathway and stimulates type I interferon (IFNs) gene expression to induce anti-tumor immunity independent of the BRCA status, providing a rationale for using PARP inhibitors as immunomodulatory agents [99,100,101]. Moreover, treatment with PARP inhibitors stimulates the type I IFN response in cells and tumors lacking BRCA2 [102]. Furthermore, PARP inhibition seems to augment cytotoxic T cell tumor infiltration through activation of the cGAS/STING innate immune pathway, leading to increased levels of chemokines, such as CXCL10 and CCL5, that induce the activation and function of cytotoxic CD8^+^ T cells [103,104]. The effect of PARP inhibition-induced T cell recruitment to tumors is more noticeable in homologous recombination-deficient compared with homologous recombination-proficient triple negative breast cancer (TNBC) cells [104].

## 7. Conclusions and Future Prospects

The promise of PARP inhibitors in cancer therapy was initially based on proposed effects on genomic integrity in the cancer cell itself. Since then, it has been uncovered that PARPs play additional roles in other important aspects of cellular biology which could be of significance for both tumor physiology and its microenvironment. Here, we can see that the immunomodulatory roles of PARP-1 and PARP-2 are complex, with specific and overlapping roles which vary by cellular compartment and context. Future work will be needed to consider how this effect of PARP inhibition on the tumor microenvironment differs by tumor type, grade, and stage. PARP inhibition may serve as an important adjuvant to immunotherapeutic strategies or indeed benefit from the checkpoint blockade itself, but will require further elucidation of the precise mechanism by which it interacts with immune pathways.

## Figures and Tables

**Figure 1 cancers-12-00392-f001:**
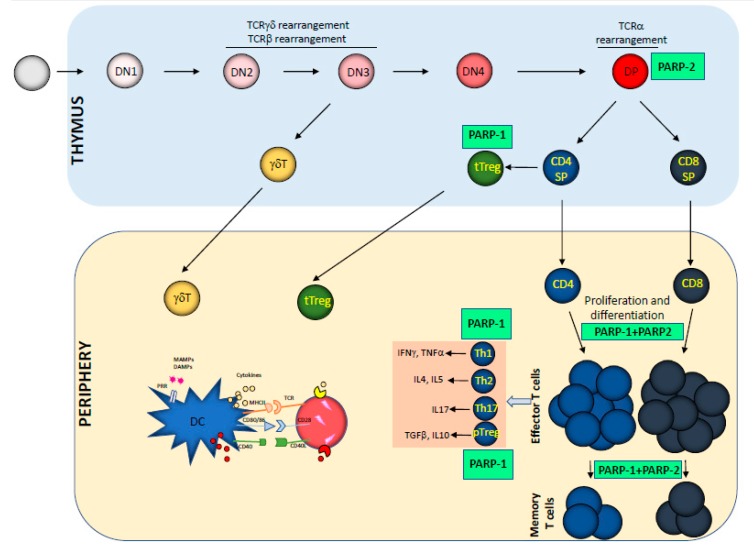
Schematic representation of T cell development depicting the specific stages in which PARP-1 and/or PARP-2 are playing a role. TCR, T cell receptor; DN, double negative; DP, double positive; SP, single positive; DC, dendritic cells; Treg, regulatory T cells.

**Figure 2 cancers-12-00392-f002:**
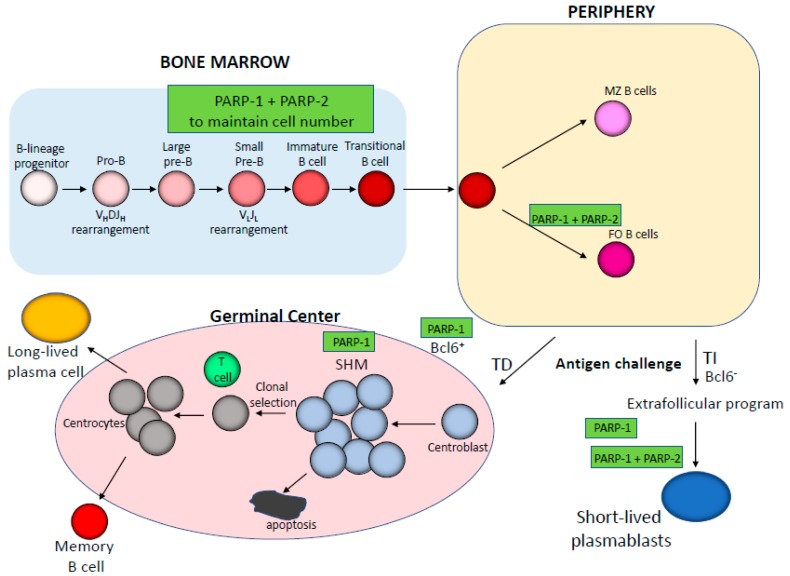
Schematic representation of B cell development depicting the specific stages in which PARP-1 and/or PARP-2 play a role. MZ, marginal zone; FO, follicular B cells; TD, T cell-dependent antigen; TI, T cell-independent antigen; SHM, somatic hypermutation.

**Figure 3 cancers-12-00392-f003:**
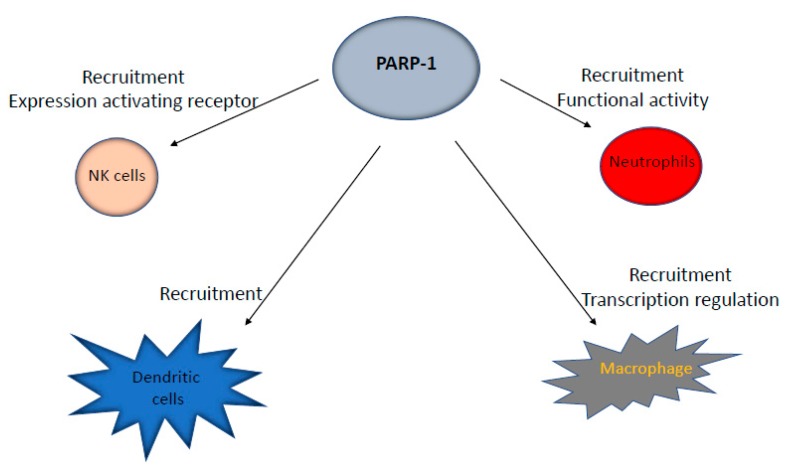
Schematic representation of the role played by PARP-1 in cells of the innate immune system.

**Figure 4 cancers-12-00392-f004:**
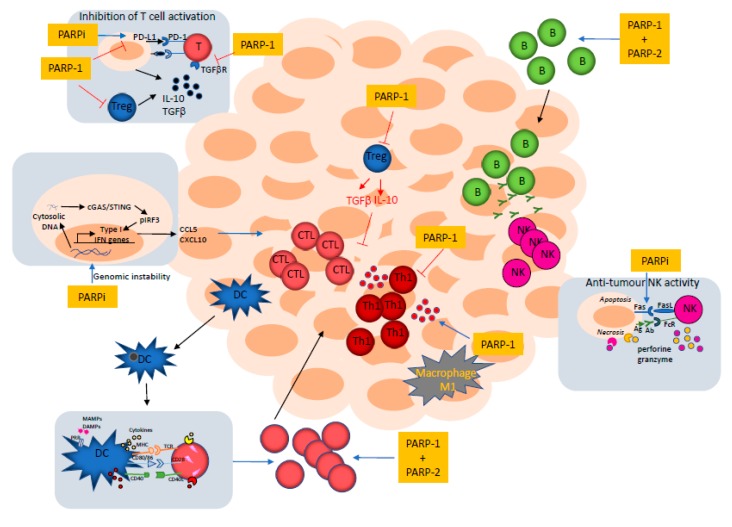
Schematic representation of the tumor microenvironment indicating the stages in which PARP-1 or its combination with PARP-2 or PARP inhibitors might play a role. In the boxes, we have included further details on the involvement of PARP-1 and/or PARP-2 or PARPi in certain contexts of the immune response. CTL, cytotoxic T cells; DC, dendritic cells; B, B cells; PARPi, PARP inhibitors.

**Table 1 cancers-12-00392-t001:** Immunological phenotypes in PARP-1- and/or PARP-2-deficient mouse models.

Immunological Process	Parp-1^-/-^	Parp-2^-/-^	Parp-1^-/-^CD4-Parp-2^f/f^	Parp-1^-/-^CD19-Parp-2^f/f^	Artd1^DMyel^	References
Thymocyte development		Decreased DP survival	Decreased DP survival			[14,15].
Peripheral T cell homeostasis			T cell lymphopenia			[17]
T cell differentiation	Bias to a Th1 phenotype Increased Treg Increased TGFβR		Compromises antibody production T cell-dependent (TD) antigens			[17] [26] [16,30,31] [31]
Central and peripheral B cell homeostasis				B cell lymphopenia		[36]
B cell function	Increased antibody production to TI antigens Enhanced somatic hypermutation(SHM)			Depletion of follicular (FO) B cells Compromises antibody production to T cell-independent (TI) antigens		[36] [36] [48]
Neutrophil function	Impaired recruitment					[52,53,54,55]
Macrophage function	Impaired recruitment				Inhibition of LPS-induced pro-inflammatory genes	[60] [63]
Dendritic cell function	Impaired recruitment					[66,67,68,69]
Natural killer (NK) cell function	Impaired recruitment					[72]

**Table 2 cancers-12-00392-t002:** Clinical trials with PARP inhibitors in combination with check-point blockade agents (www.clinicaltrials.gov).

PARPi	IMMUNE CHECK-POINT INHIBITOR	CLINICAL PHASE	CONDITIONS	IDENTIFIER
Talazoparib	Avelumab (anti-PD-L1)	II	Advanced or metastatic solid tumors	NCT03330405
Pamiparib	Tislelizumab (anti-PD-1)	I/Ib	Solid tumors	NCT02660034
Rucaparib	Nivolumab (anti-PD-1)	Ib/IIa	Prostate Cancer, Endometrial Cancer	NCT03572478
Olaparib	Tremelimumab (anti-CTLA-4)	I/II	Ovarian Cancer, Fallopian Tube Cancer, Peritoneal Neoplasms	NCT02571725
Talazoparib	Avelumab (anti-PD-L1)	III	Ovarian Cancer	NCT03642132
Rucaparib	Nivolumab (anti-PD-1)	II	Biliary Tract Cancer	NCT03639935
Niraparib	PD-1 Inhibitor	II	Lung Neoplasms	NCT03308942
Talazoparib	Pembrolizumab (anti-PD1)	I/II	Solid Tumor, Epithelial Ovarian Cancer, Fallopian Tube Cancer, Peritoneal Cancer, Triple Negative Breast Cancer, Small Cell Lung Cancer, Metastatic Breast Cancer, Malignant Melanoma, Non-Small Cell Lung Cancer, Urothelial Carcinoma	NCT04158336
Olaparib	Atezolizumab (anti-PD-L1)	II	Locally Advanced Unresectable Breast Carcinoma, Metastatic Breast Carcinoma, Stage III Breast Cancer AJCC v7, Stage IIIA Breast Cancer AJCC v7, Stage IIIB Breast Cancer AJCC v7, Stage IIIC Breast Cancer AJCC v7, Stage IV Breast Cancer AJCC v6 and v7	NCT02849496
Olaparib	Durvalumab (Anti-PD-L1)	II	Endometrial Neoplasms, Uterine Neoplasms, Endometrium Cancer	NCT03951415
Talazoparib	Avelumab (anti-PD-L1)	I/II	Breast Cancer	NCT03964532
Olaparib	Durvalumab (Anti-PD-L1)	II	Mismatch Repair Proficient Colorectal Cancer, Pancreatic Adenocarcinoma, Leiomyosarcoma	NCT03851614
Olaparib	Durvalumab (Anti-PD-L1)	II	Triple Negative Breast Cancer	NCT03167619
Olaparib	Durvalumab (Anti-PD-L1)	I	Anatomic Stage IV Breast Cancer AJCC v8, Estrogen Receptor Negative, HER2/Neu Negative, Progesterone Receptor Negative, Prognostic Stage IV Breast Cancer AJCC v8, Triple-Negative Breast Carcinoma	NCT03544125
Niraparib	Dostarlimab (Anti-PD-1)	II/III	Ovarian Carcinosarcoma, Endometrial Carcinosarcoma	NCT03651206
Veliparib	Nivolumab (anti-PD-1)	I	Advanced Solid Neoplasm, Aggressive Non-Hodgkin Lymphoma, Recurrent Solid Neoplasm, Refractory Mantle Cell Lymphoma, T-Cell Non-Hodgkin Lymphoma, Unresectable Solid Neoplasm	NCT03061188
Rucaparib	Nivolumab (anti-PD-1)	II	Epithelial Ovarian Cancer, Fallopian Tube Cancer, Primary Peritoneal Carcinoma, High Grade Serous Carcinoma, Endometrioid Adenocarcinoma	NCT03824704
Talazoparib	Avelumab (anti-PD-L1)	II	Squamous Cell Carcinoma of the Head and Neck (SCCHN), Metastatic Castration Resistant Prostate Cancer (mCRPC)	NCT04052204
Niraparib	Pembrolizumab (anti-PD1)	I/II	Triple Negative Breast Cancer, Ovarian Cancer, Breast Cancer, Metastatic Breast Cancer, Advanced Breast Cancer, Stage IV Breast Cancer, Fallopian Tube Cancer, Peritoneal Cancer	NCT02657889
Olaparib	Durvalumab (anti-PD-L1)	II	Metastatic Triple Negative Breast Cancer, Breast Cancer, ER-Negative PR-Negative HER2-Negative Breast Cancer, ER-Negative PR-Negative HER2-Negative Breast Neoplasms, Triple-Negative Breast Cancer, Triple-Negative Breast Neoplasm	NCT03801369
Olaparib	Durvalumab (anti-PD-L1)	II	Prostate Cancer	NCT03810105
Rucaparib	Nivolumab (anti-PD1)	II	Small Cell Lung Cancer	NCT03958045
veliparib	Nivolumab (anti-PD1)	I	Non-Small Cell Lung Cancer	NCT02944396
Olaparib	Pembrolizumab (anti-PD1)	III	Prostatic Neoplasms	NCT03834519
Olaparib	Durvalumab (anti-PD-L1) and Tremelimumab (anti-CTLA-4)	II	BRCA1 Gene Mutation, BRCA2 Gene Mutation, Ovarian Serous Adenocarcinoma, Recurrent Fallopian Tube Carcinoma, Recurrent Ovarian Carcinoma, Recurrent Primary Peritoneal Carcinoma	NCT02953457
Olaparib	Durvalumab (anti-PD-L1)	II	Squamous Cell Carcinoma of the Head and Neck	NCT02882308
Olaparib	Durvalumab (anti-PD-L1)	II	Glioma, Cholangiocarcinoma, Solid Tumor, IDH Mutation	NCT03991832
Niraparib	Dostarlimab (anti-PD1)	III	Ovarian Cancer	NCT03602859
Olaparib	Durvalumab (anti-PD-L1)	I	Advanced Malignant Solid Neoplasm, Metastatic Malignant Solid Neoplasm, Unresectable Malignant Solid Neoplasm	NCT03842228
Niraparib	Nivolumab (anti-PD1) or Ipilimumab (anti-CTLA4)	I II	Pancreatic Adenocarcinoma	NCT03404960
Niraparib	Atezolizumab (anti-PD-L1)	II	Solid Tumor	NCT04185831

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
