# Peer review of "Immunomodulatory Roles of PARP-1 and PARP-2: Impact on PARP-Centered Cancer Therapies"

_cancers, 2020, doi:10.3390/cancers12020392_

Round 1

Reviewer 1 Report

In this manuscript, the authors exhaustively review the diverse and pleiotropic roles PARP1 and PARP2 play in immune cell development and function, implying that PARP inhibitor drugs not only target cancer cells but also impact on immune responses against tumors. They authors discuss how this can be exploited to improve anticancer therapeutic strategies.

The review is about a trending topic, still approaching it by a quite original perspective. The text is clearly written and the figures effectively illustrate the major points. To the best of my knowledge, all the more relevant data about the treated topics have been cited and discussed.

The authors should, though, carefully check the reference list before publication: often the style does not match Cancers guidelines.

Author Response

Please, find enclose our response

Reviewer 2 Report

This manuscript reviews the roles of PARP-1 and PARP-2 in modulation of immune system development and activity. The authors concentrate on the phenotypes of PARP-1 and/or PARP-2 deficient mice and try to extrapolate this to a medical setting of PARP inhibitor treatment.

This review is rather one-sided in its concentration on genetic PARP depletion. There is quite some literature on the effects of PARP inhibitors, which should complement this story. This might also prevent too much emphasis on single and double knock out mouse models (that have been studied in the lab of the first author), making the review rather lab-centered in its current form. This is also evidenced by statements like ‘As our lab showed’.

In addition to this general remark on the content of the manuscript, it would also be useful if the authors could be a bit more precise in their description of the major findings in the papers they reference. For example, I miss a critical evaluation of PARP-1 deficient mice with T- or B-cell specific knock out of PARP-2. All effects are currently interpreted as specific functions of PARP, but it is also possible that the complete knock out is not viable at the single-cell level at some stage of development and could be a rather non-specific effect of a general lethal effect of the double knock out. It is very difficult to translate this to transient PARP inhibition during cancer treatment. A better evaluation of this difference would help the manuscript.

Line 112-113 speaks of ‘synthetic lethality’, but this is more an example of ‘redundancy’ of two proteins that can take over each other’s function. Synthetic lethality is if two processes can be affected one-by-one without major effects, but not both at the same time.

Line 243-246: the reference does not describe support this statement. Is this unpublished work from the own lab? If so, this should be referenced correctly. It would also require some more detail to make it believable.

Figure 4 is extremely complicated. I would suggest to simplify this for a paper.

Line 274-277: This paper shows that only PARP-1 protein is required, not its activity. This should clearly be explained when extrapolating this to a setting of PARP inhibition.

Lines 299-307: I miss the connection to PARP inhibition here.

Lines 333-346: I miss a description of the (rather disappointing) results of clinical studies that combine PARP inhibitors with immune checkpoint inhibitors. There are promising data in mouse models, but the first clinical studies do not look promising at all. This should at least be mentioned.

Furthermore, language editing would improve the manuscript considerably. In several places there are missing particles (e.g. ‘at posttranslational level’ instead of ‘at the posttranslational level’), typos (immunomological instead of immunological) and word endings (the previous mentioned instead of the previously mentioned).

Author Response

Please, find enclose our response

Round 2

Reviewer 2 Report

Most issues have been addressed in the revised version. I am still not convinced that PARP knock out is a good model for predicting responses to PARP inhibition, but this is at least discussed to some extent and references have been added. I think the review could have been a bit more balanced, but it is at least not misleading or completely one-sided.